# Routine health information utilization and associated factors among health care workers in Ethiopia: A systematic review and meta-analysis

Birye Dessalegn Mekonnen[1]*, Senafekesh Biruk Gebeyehu[2]

**1** Department of Nursing, Teda Health Science College, Gondar, Ethiopia, **2** Department of Health Informatics, Teda Health Science College, Gondar, Ethiopia

* birye22@gmail.com

## Abstract

### Background

Utilization of routine health information plays a vital role for the effectiveness of routine and programed decisions. A proper utilization of routine health information helps to make decisions based on evidence. Considerable studies have been done on the utilization of routine health information among health workers in Ethiopia, but inconsistent findings were reported. Thus, this study was conducted to determine the pooled utilization of routine health information and to identify associated factors among health workers in Ethiopia.

### Methods

Search of PubMed, HINARI, Global Health, Scopus, EMBASE, web of science, and Google Scholar was conducted to identify relevant studies from October 24, 2020 to November 18, 2020. The Newcastle-Ottawa scale tool was used to assess the quality of included studies. Two reviewers extracted the data independently using a standardized data extraction format and exported to STATA software version 11 for meta-analysis. Heterogeneity among studies was checked using Cochrane Q and $I^2$ test statistics. The pooled estimate of utilization of routine health information was executed using a random effect model.

### Results

After reviewing 22924 studies, 10 studies involving 4054 health workers were included for this review and meta-analysis. The pooled estimate of routine health information utilization among health workers in Ethiopia was 57.42% (95% CI: 41.48, 73.36). Supportive supervision (AOR = 2.25; 95% CI: 1.80, 2.82), regular feedback (AOR = 2.86; 95% CI: 1.60, 5.12), availability of standard guideline (AOR = 2.53; 95% CI: 1.80, 3.58), data management knowledge (AOR = 3.04; 95% CI: 1.75, 5.29) and training on health information (AOR = 3.45; 95% CI: 1.96, 6.07) were identified factors associated with utilization of routine health information.

**Data Availability Statement:** All relevant data are within the paper and its Supporting Information files.

**Funding:** The authors received no specific funding for this work.

**Competing interests:** The authors have declared that no competing interests exist.

**Abbreviations:** AOR, Adjusted odd ratio; CI, Confidence intervals; PRISMA, Preferred Reporting Items for Systematic review and Meta-analysis; SNNPR, South Nation and Nationality Peoples Regional.

## Conclusion

This systematic review and meta-analysis found that more than two-fifth of health workers did not use their routine health information. This study suggests the need to conduct regular supportive supervision, provision of training and capacity building, mentoring on competence of routine health information tasks, and strengthening regular feedback at all health facilities. In addition, improving the accessibility and availability of standard set of indicators is important to scale-up information use.

## Background

Health information is the processed and generated data that an individual, group or institution use to support their decisions in the health care system [1]. It is essential for the entire health system by providing the right information for evidence-based health practices and improving managerial decisions [2,3].

Routine health information utilization is vital for the day-to-day patient management, disease prioritization, health education, resource allocation, and decision making as well as for the planning, monitoring, and evaluation of health care service activities [4]. A properly functioning of routine health information system helps to get the right information at the right time into the right hands, which supporting policymakers, managers, and service providers to make decisions based on evidence [5,6].

In developing countries, the utilization of routine data for decision making remains very weak mainly due to inadequate data analysis and health information systems [7,8]. Though most health care providers report routine health data, understanding the benefits of routine health information and utilization remains low in low income countries [9–11]. As a result, data usually sat on shelves, cabinets without sufficiently processed and utilized for program and policy improvements [12,13]. This leads to challenges and difficulties to the efficiency and effectiveness of health care delivery [14].

In Ethiopia, Information Revolution is one of the four transformation agendas in the Health Sector Transformation Plan (HSTP), which involves an important shift from old methods of information utilization to practical use of information [15,16]. Information Revolution, as transformation agenda sets a priority for the generation and utilization of health information [17]. In addition, the Ethiopian federal ministry of health has been incorporating new initiatives which are more comprehensive and focused on strengthening the standardization process [16]. Even with these efforts, the utilization of routine health information in Ethiopia is still a big challenge [18,19].

The identified factors that prevent utilization of routine health information including, analysis skills, organizational infrastructure and training, lack of culture of information use, lack of supervision and regular feedback, availability of human resources, knowledge, computer skill, work load, computer access, availability of guidelines and formats, and data quality [5,9,20–22]. Furthermore, a limited use of routine health data was observed in health care workers who lack training on computer software and data management [22,23].

Despite the fact utilization of routine health information is important for operational, tactical and strategic decision-making, poor data quality and limited use remain the major concerns [16]. Thus, for the effective intervention of routine health information utilization and its factors, determination of the level of utilization and identifications of associated factors is important.

In this study, literature on utilization of routine health information among health workers in Ethiopia were reviewed. However, the studies show a difference in routine health information utilization and associated factors, and to the authors knowledge, the literatures have not been examined systematically. Therefore, this systematic review and meta-analysis was aimed to estimate the pooled utilization of routine health information and to identify associated factors among health workers in Ethiopia. The findings of this meta-analysis will help for policy-makers, and other stakeholders to effectively implement different health sector strategies and programs, social and community health insurances, and health care financing. The finding from this study will also help health workers to design suitable intervention to improve evidence-based practice and to understand their routine health information utilization level.

## Methods

This systematic review and meta-analysis was prepared and presented based on Preferred Reporting Items for Systematic Reviews and Meta-Analyses (PRISMA) checklist (S1 Checklist).

### Eligibility criteria

Original research studies reporting the utilization of routine health information and/or associated factors among health workers in Ethiopia were included in the study. Observational studies with no restrictions on publication year were considered. Both published and unpublished articles, but written only in English language were considered for inclusion. All publications reported up to November 18, 2020 were considered.

Studies that did not clearly report the utilization of routine health information among health workers in Ethiopia were excluded. In addition, articles without full text, and abstract, editorial reports, letters, reviews, and commentaries were excluded from the study.

### Search strategy and information sources

A comprehensive and systematic search of literature was carried out through electronic databases including PubMed, HINARI, Global Health, Scopus, EMBASE, web of science, African journal online (AJOL), and Google Scholar from October 24, 2020 to November 18, 2020. The search was done using the following keywords and Medical Subject Headings (MeSH) terms: "utilization" OR "practice" AND "health communication" OR "health information" AND "associated factors" OR "determinants" AND "health professionals" OR "health care workers" OR "healthcare facilities" AND "Ethiopia". The search focused on studies with epidemiological data on the utilization of routine health information and associated factors among health workers in Ethiopia.

**Data extraction.** After screening of titles, abstracts and the full texts of each included original studies, data were extracted using a standardized data extraction tool which was adapted from the Joanna Briggs Institute (JBI). Two reviewers (BDM & SBG) extracted the data independently, and reviewed all the included articles. Any disagreement between reviewers were resolved through discussion.

The study characteristics, such as the name of first author, study region and setting, year of publication, study design, study participants, sampling technique, data source, sample size, and response rate were extracted. Prevalence (utilization) of routine health information and risk factors with 95% confidence intervals were also extracted.

## Risk of bias (quality) assessment

The quality of each original studies was assessed by using the Newcastle-Ottawa scale (NOS) tool adapted for cross-sectional studies quality assessments. The assessment tool contains three main parts. The first part of the tool has five-stars, and assesses the methodological quality of each study (i.e., sampling technique, sample size, response rate, and ascertainment of the risk factor or exposure). The second part of the tool assesses the comparability of the study with a possibility of two stars to be gained. The last component of the instrument measures the outcomes and statistical tests of the primary study with a possibility of three stars to be gained. Finally, studies included in this systematic review and meta-analysis have medium (5–6 out of 10 stars) to high quality scores (>6 out of 10 stars). Two authors (BDM & SBG) independently assessed the quality of studies included in the review. Disagreements between reviewers during quality assessment were handled through discussion.

## Outcome measurement

The primary outcome measure of this meta-analysis and systematic review is utilization of routine health information. Utilization of routine health information was assessed using the Performance of Routine Information System Management (PRISM) assessment tool. It was defined as the use of routine health information for monitoring day to day health service activities, developing weekly plan, service delivery improvement, displaying updated information, drug procurement, resource mobilization, facilitating community mobilization, detecting the cause of health problem in the community, prediction of outbreaks, and disease prioritization. For this study, utilization of routine health information was computed by dividing the number of health workers who had a good level of routine health information utilization by the total number of health workers in the study (sample size) multiplied by 100. The second outcome variable of the study was to identify factors associated with utilization of routine health information among health workers in Ethiopia, which were measured in the form of the odds ratio (OR). Odds ratio was calculated for each identified factor based on the binary outcome data reported by each primary study.

## Data synthesis and analysis

The data were extracted using a Microsoft Excel spreadsheet and then imported into STATA version 14 for further analysis. The primary studies were described and summarized using tables, figures, and forest plots. The pooled estimate of utilization of routine health information was executed using a random effect model with 95% confidence interval (CI). The measure of association for factors that determine utilization of routine health information among health workers was estimated using odds ratio with 95% CI. Random effect model was computed during meta-analysis as heterogeneity was exhibited among the included studies. Heterogeneity among the recorded prevalence of studies has been assessed with Cochran's Q statistic and the $I^2$ statistics. Furthermore, subgroup analysis was done to reduce the random variations among the point estimates of the primary studies. Visual inspection of asymmetry in funnel plots, and Egger regression tests were employed to assess the existence of publication bias.

## Result

## Search results

A total of 22924 articles regarding the utilization of routine health information and/or associated factors among health workers in Ethiopia were retrieved. Among the total retrieved

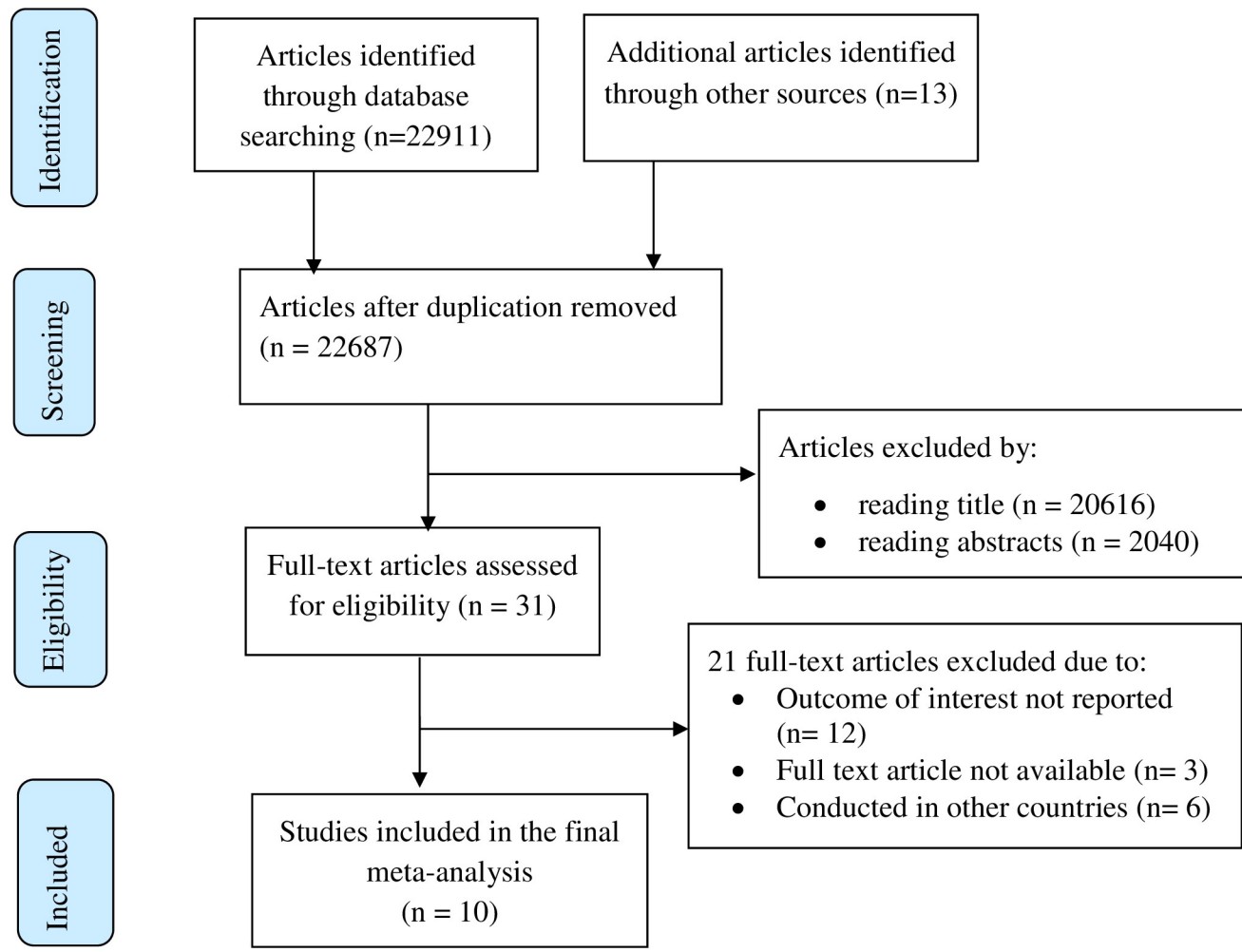

**Fig 1. Flow chart of study selection for systematic review and meta-analysis of utilization of routine health information and associated factors among health workers in Ethiopia, 2020.**

studies, 237 studies were removed due to duplication. After assessing the articles based on their titles and abstracts, 22656 articles were excluded. The remaining 31 full-text articles were assessed for eligibility criteria, which resulted in the further exclusion of 21 articles mainly because of variation in the study population and unreported outcome of interest. As a result, 10 studies were included to undergo the final meta-analysis (Fig 1).

## Characteristics of the included studies

All included studies used facility based cross-sectional study design to estimate utilization of routine health information. All the studies included for this review were published from 2011 up to 2020. Five of the studies included in this review used purposive sampling technique, two used systematic random sampling, two used cluster sampling technique, and one used multi-stage sampling technique. Of the included studies, four studies used self-administered and observation, four studies used only self-administered, and two studies used interviewer-administered method to select study participants. From an estimated 4144 health care workers, a total of 4054 participants were involved with an estimated sample size range from 239 [24] up to720 [21]. The included studies reported that the utilization of routine health information

**Table 1. Descriptive summary of primary studies included in the meta-analysis of utilization of routine health information and associated factors among health workers in Ethiopia, 2020.**

| First author, publication year | Region | Study area | Study design | Study population | Sampling technique | Data collection method | Sample size | Response rate (%) | Prevalence (%) |
|---|---|---|---|---|---|---|---|---|---|
| Asemahagn MA [27], 2017 | Amhara | East Gojjam zone | IBCSS | Health Center and unit heads | Systematic random sampling | Self-administered and observation | 250 | 100 | 38.4 |
| Shiferaw AM et al [26], 2017 | Amhara | East Gojjam zone | IBCSS | Health workers | Cluster sampling technique | Self-administered | 668 | 97.8 | 45.7 |
| Dagnew E et al [21], 2018 | Amhara | North Gondar | IBCSS | Health care professional | Multi-stage sampling technique | Self-administered and observation | 720 | 100 | 78.5 |
| Wude H et al [29], 2020 | SNNPR | Hadiya zone | IBCSS | Health workers | Systematic random sampling | Self-administered | 490 | 98 | 62.7 |
| Yitayew S et al [28], 2019 | Amhara | East Gojjam zone | IBCSS | Health extension workers | Purposive sampling | Self-administered | 302 | 100 | 53.3 |
| Abajebel S et al [25], 2011 | Oromia | Jimma Zone | FBCSS | Health facility and unit heads | Purposive sampling | Observation and interview | 362 | 100 | 32.9 |
| Andualem M et al [9], 2013 | Amhara | Bahir dar Town | IBCSS | Health workers | Purposive sampling | Self-administered and observation | 350 | 96.9 | 97.1 |
| Shagak S et al [30], 2014 | SNNPR | Gamo Gofa Zone | IBCSS | Health extension workers | Cluster sampling technique | Self-administered | 457 | 92.1 | 58.2 |
| Teklegiorgis K et al [24], 2014 | Dire Dawa | Dire Dawa | FBCSS | Health facility and unit heads | Purposive sampling | Self-administered | 239 | 100 | 52.7 |
| Emiru K et al [31], 2018 | Oromia | East Wollega zone | IBCSS | Health facility and unit heads | Purposive sampling | Face to face interview | 306 | 100 | 54.2 |

among health workers ranged from 32.9% [25] to 97.1% [9]. Five of the studies included in this review were conducted from Amhara region [9,21,26–28], two were from Southern Nations Nationalities and People's Region (SNNPR) [29,30], two were from Oromia region [25,31], and the remaining one was from Dire Dawa administrative city [24] (Table 1).

## Meta-analysis

**Risk of bias assessment for the included studies.** The quality of each original studies was critically assessed using the Newcastle-Ottawa scale tool adapted for cross-sectional studies. From the total included studies, the quality assessment summary showed about four-fifth (n = 8, 80%) of the studies had high quality, and the reaming one-fifth (n = 2, 20%) of studies had medium quality (Table 2).

**Table 2. Quality assessment of primary studies included in the meta-analysis of utilization of routine health information and associated factors among health workers in Ethiopia, 2020.**

| Studies ID | Selection (Maximum of five star) | Comparability (Maximum two star) | Outcome assessment (Maximum of three stars) | Overall quality |
|---|---|---|---|---|
| Asemahagn MA | **** | * | *** | High |
| Shiferaw AM et al | **** | ** | ** | High |
| Dagnew E et al | ***** | ** | ** | High |
| Wude H et al | **** | ** | *** | High |
| Yitayew S et al | *** | * | ** | Medium |
| Abajebel S et al | *** | * | *** | High |
| Andualem M et al | **** | * | ** | High |
| Shagak S et al | **** | * | ** | High |
| Teklegiorgis K et al | *** | * | ** | Medium |
| Emiru K et al | **** | ** | *** | High |

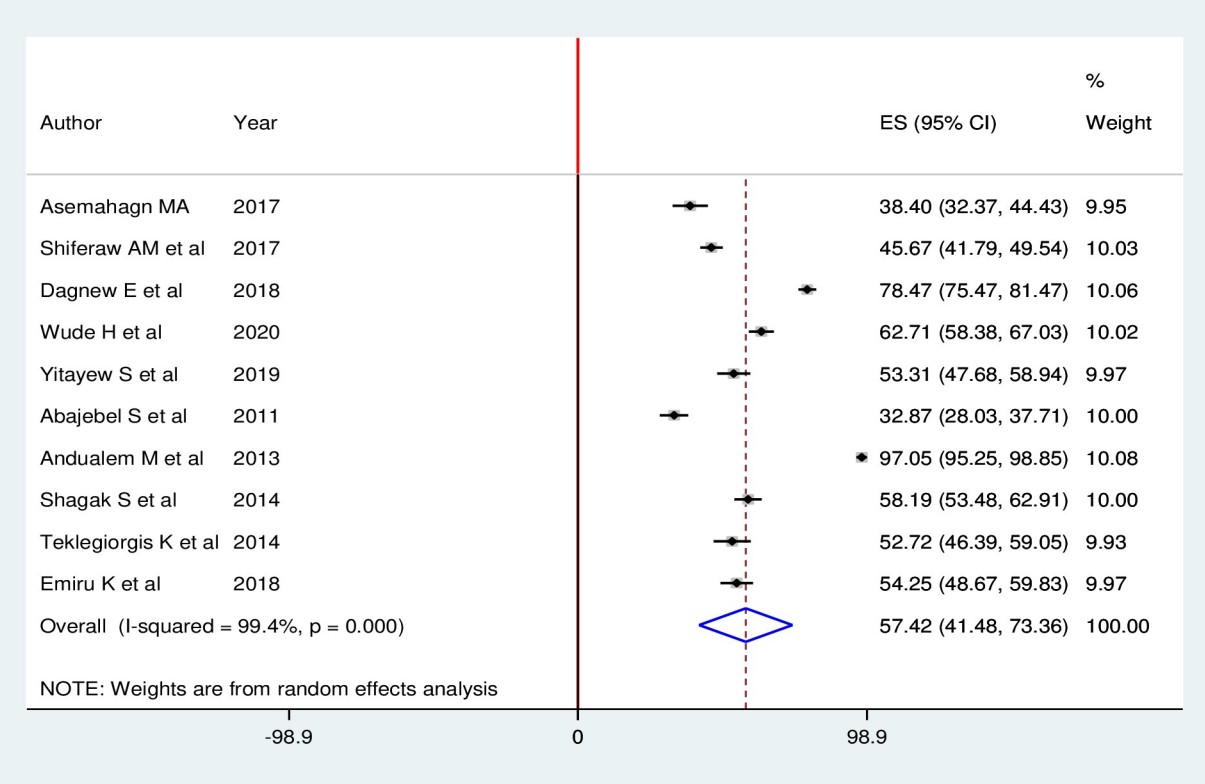

**Fig 2. Forest plot of the pooled utilization of routine health information among health workers in Ethiopia, 2020.**

**Utilization of routine health information among health workers in Ethiopia.** Overall, the pooled estimate of routine health information utilization among health workers in Ethiopia was 57.42% (95% CI: 41.48, 73.36). High heterogeneity across the included studies was exhibited ($I^2$ = 99.4%; p < 0.001) in estimating the pooled utilization of routine health information among health workers. Hence, to estimate the pooled prevalence of routine health information utilization among health workers, random effects model was used during meta-analysis (Fig 2).

**Subgroup analysis.** Subgroup analysis was carried out based on the regions where the primary studies were conducted. Accordingly, the highest routine health information utilization was observed in Amhara region with a prevalence of 62.67%(95% CI: 39.36, 85.97), and the lowest routine health information utilization was observed in Oromia region with a prevalence of 43.51%(95% CI: 22.57, 64.46) (Fig 3).

**Publication bias.** Visual inspection of the asymmetry in funnel plots, and Egger regression tests were employed to assess the existence of publication bias. Accordingly, the result of both funnel plots and the Egger's tests revealed the absence of publication bias in the included studies. The result of Egger's test was not statistically significant (p = 0.182), which declared absence of publication bias. Additionally, visual inspection of the funnel plots showed a symmetric distribution of studies (Fig 4).

**Factors associated with utilization of routine health information.** In this study, some of the factors associated with utilization of routine health information were pooled quantitatively and some were not due to inconsistent classification (grouping) of the independent variables with respect to the outcome (utilization of routine health information).

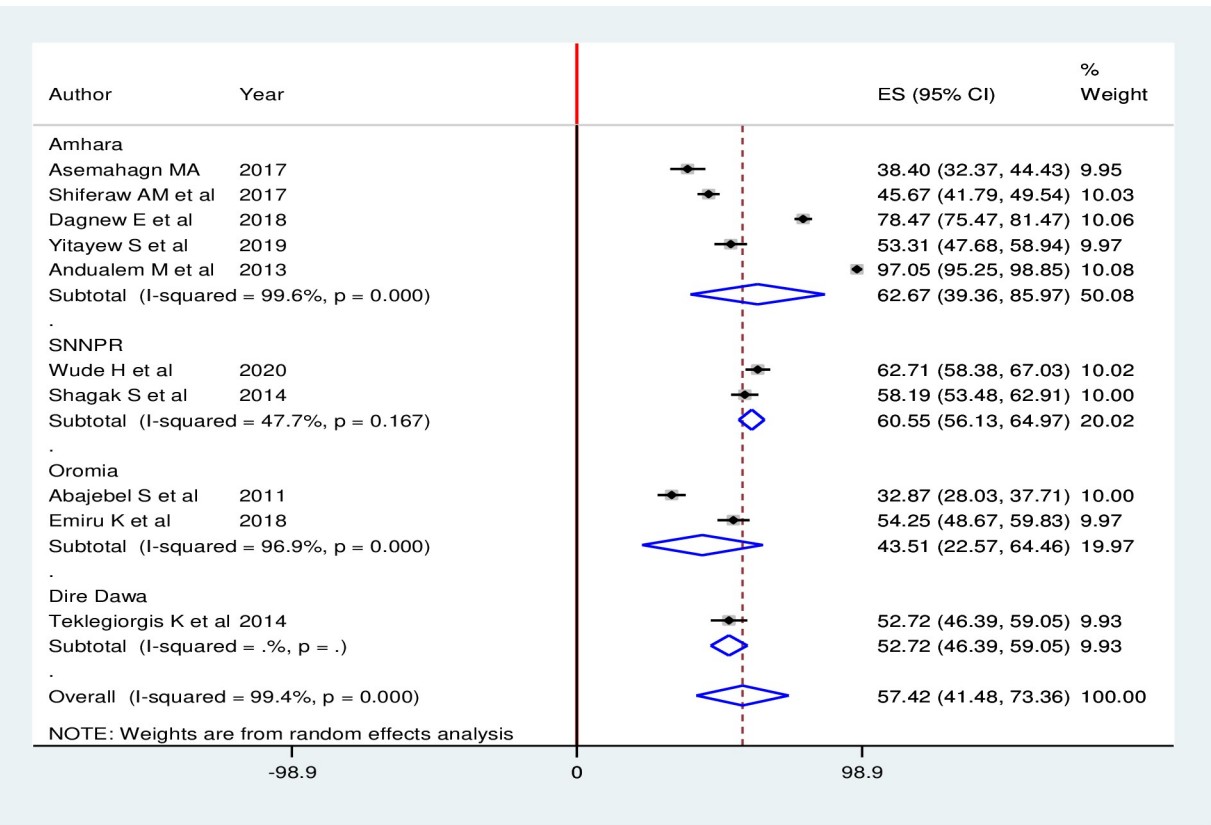

**Fig 3. Subgroup analysis (by region) of studies included in meta-analysis on utilization of routine health information among health workers in Ethiopia, 2020.**

Six studies indicated that supportive supervision has a significant association with utilization of routine health information. The odds of routine health information utilization were 2.25 times (AOR = 2.25; 95% CI: 1.80, 2.82) higher among health care workers who received supportive supervision on routine health information when compared with those who have not received supervision. In this meta-analysis, included studies were characterized with no existence of heterogeneity ($I^2$ = 0.0%, P = 0.939). Thus, a fixed effect model analysis was used (Fig 5).

Five studies showed that regular feedback has a significant association with utilization of routine health information. Health workers who got regular feedback were 2.86 times (AOR = 2.86; 95% CI: 1.60, 5.12) more likely to use routine health information than those who did not get feedback. A random effect model was used in this meta-analysis as the included studies were characterized by existence of heterogeneity ($I^2$ = 84.3%, P <0.001) (Fig 6).

Three studies also showed that availability of standard guideline has an association with utilization of routine health information. The odds of utilization of routine health information were about 2.53 times (AOR = 2.53; 95% CI: 1.80, 3.58) higher among health workers who have standard guideline than their counterpart. A fixed effect model was used in this meta-analysis as the included studies were characterized by absence of heterogeneity ($I^2$ = 0.0%, P = 0.463) (Fig 7).

Furthermore, four studies indicated that data management knowledge has an association with utilization of routine health information. Accordingly, increased odds of good health information use were observed among health care workers who had good data management

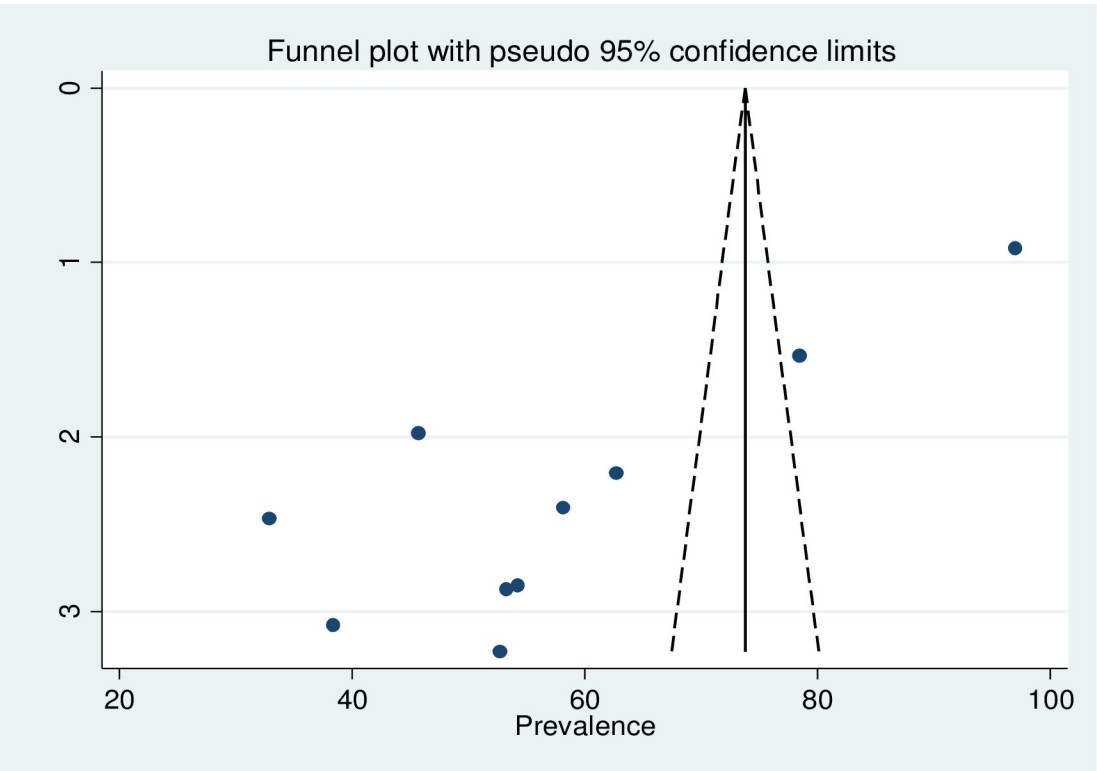

**Fig 4. Graphic representation of publication bias using funnel plots of all included studies, 2020.**

knowledge than their counterpart (AOR = 3.04; 95% CI: 1.75, 5.29). A random effect model was used in this meta-analysis as the included studies were characterized by existence of heterogeneity ($I^2$ = 76.1%, P = 0.006) (Fig 8).

Four studies indicated that training on health information has an association with utilization of routine health information. The odds of utilization of routine health information were about 3.45 times (AOR = 3.45; 95% CI: 1.96, 6.07) higher among trained health workers when compared with their counterparts. A random effect model was used in this meta-analysis as the included studies were characterized by existence of low heterogeneity ($I^2$ = 69.1%, P = 0.021) (Fig 9).

## Discussion

Use of routine health information can potentially circumvent several of the structural and systemic barriers faced by health workers in delivering health care. Evidence suggests that use of routine health information for healthcare delivery is feasible for health workers irrespective of their education or prior training [32]. Thus, this systematic review and meta-analysis was conducted to estimate the pooled prevalence of routine health information utilization and associated factors among health workers in Ethiopia. Accordingly, more than two-fifth of health workers did not use their routine health information. This finding implies that the need for close monitoring and evaluation of the strategies that promoted the utilization of data generated from health care systems. Furthermore, this finding infers the need to make plans once identify performance gaps. Evidence revealed that strengthening health information system focusing on organizational structures, technical, and behavioral is one important components for improving the quality and use of data for decision making [33,34].

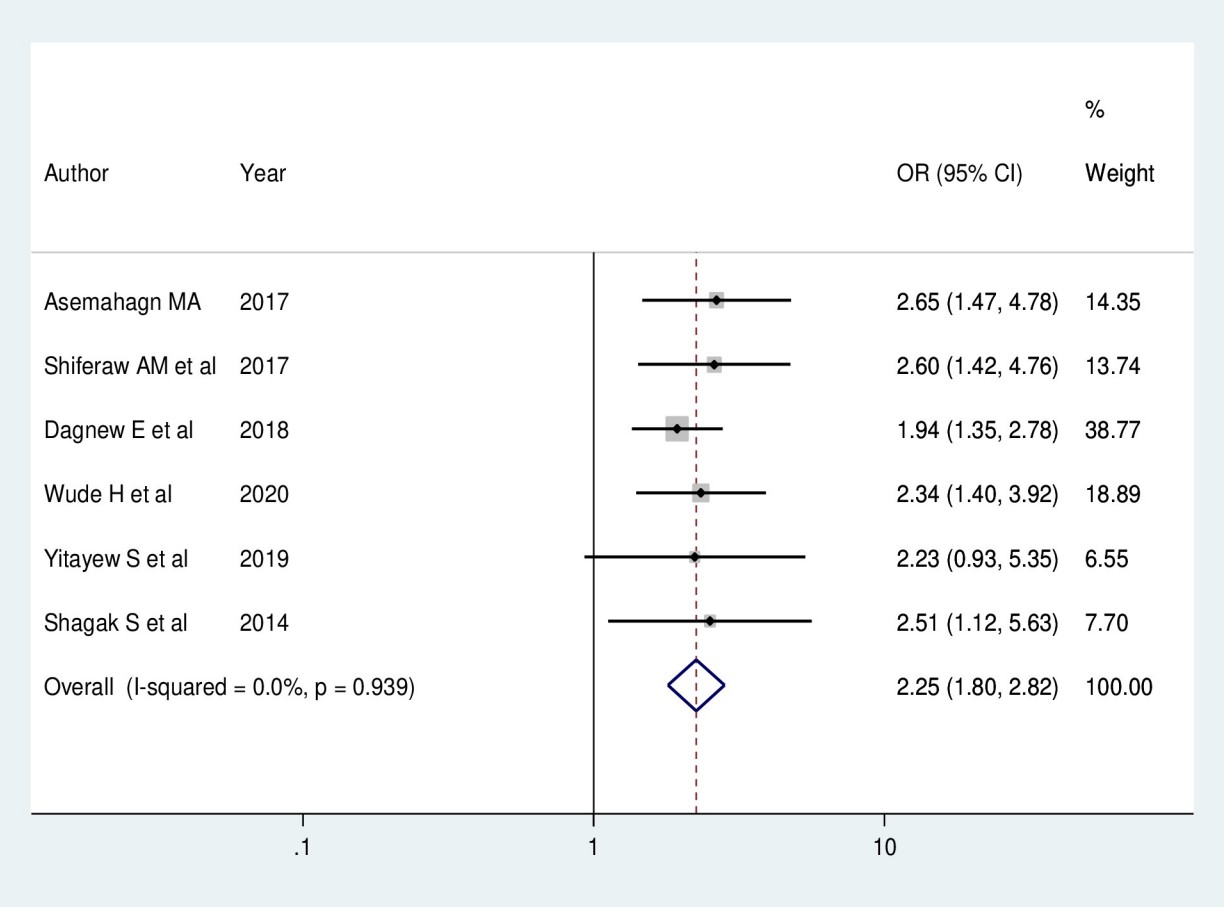

**Fig 5. Forest plot showing the pooled odds ratio of the association between supportive supervision and utilization of routine health information health workers in Ethiopia, 2020.**

In this review, the pooled estimate of routine health information utilization among health workers in Ethiopia was 57.42% (95% CI: 41.48, 73.36). Even though there is no meta-analysis on this research question, the utilization of routine health information reported in the present study is consistent with other studies conducted in Uganda (59%) [35], Tanzania (58%) [36] and South Africa, (65%) [37]. However, the finding in this meta-analysis is higher than a study carried out in Cote D'Ivoire which reported the utilization of routine health information as 38% [11]. This variation might be due to differences in health information system structures and health care workers' attitude towards routine health information utilization [38].

This meta-analysis revealed that health workers who received supportive supervision were more likely to use routine health information as compared with those who have not received supervision. This finding was supported by other studies conducted in Tanzania [36] and rural of South Africa [37]. This could be due to the fact that supportive supervision has an important role in identifying organizational, technical and behavioural gaps, and improving health workers' performance.

This study also indicated that health workers who got regular feedback were 2.86 times more likely to use routine health information than those who did not get feedback. This finding implies the need to give due attention for all levels of health facilities in terms of regular feedback by the government [39]. Literatures also documented that regular feedback given to

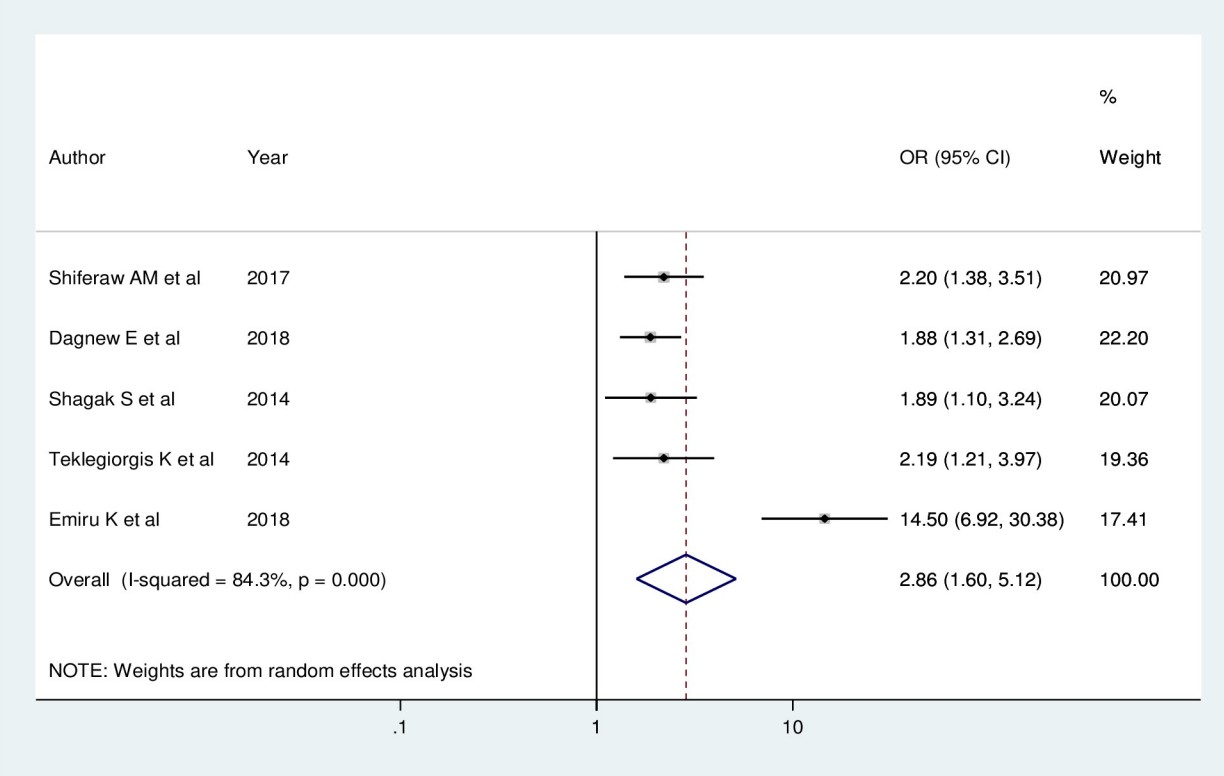

**Fig 6. Forest plot showing the pooled odds ratio of the association between regular feedback and utilization of routine health information health workers in Ethiopia, 2020.**

health workers is important to improve the utilization of routine information in health care systems [39,40].

Availability of standard guideline was another determinants of routine health information utilization. Accordingly, the odds of utilization of routine health information were about 2.53 times higher among health workers who have standard guideline than their counterpart. This finding is supported by other studies. This could be due to the fact that the presence of data sources (standard indicators, guidelines) can help health workers to utilize routine health information for evidence based decision making [41].

This study also identified that data management knowledge has an association with utilization of routine health information. Health workers who had good data management knowledge were more likely to use routine health information as compared with their counterpart. This finding is supported by other previous studies [23,42,43]. This could be due to the fact that health workers with adequate knowledge on how to process and manage health information can develop skills in their daily activities, so that they can use routine health information easily. Furthermore, health workers who have good data management knowledge can transform data into meaningful information for utilizing routine health information. An evidence from India revealed that utilization of health information depends on data analysis skills and organizational factors [44].

Furthermore, training on health information was another determinants of routine health information utilization. The odds of utilization of routine health information were about 3.45 times higher among trained health workers when compared with their counterparts. This finding is supported by other studies [23,42]. This could be due to the fact that health workers who

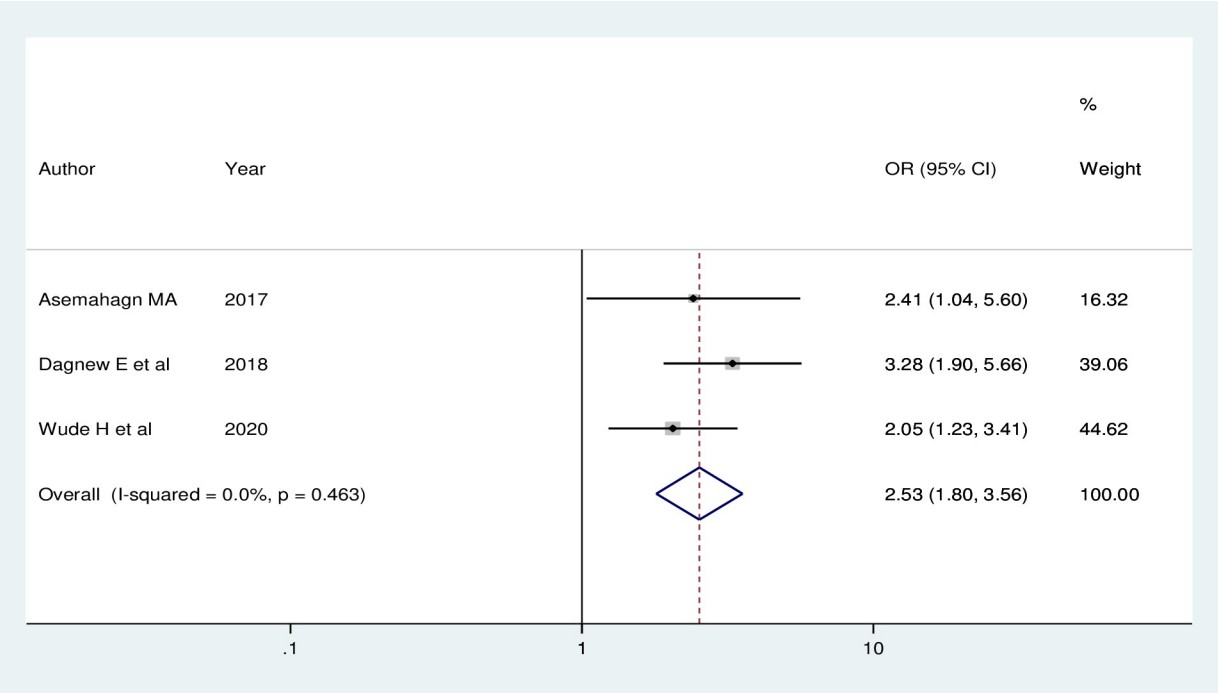

**Fig 7. Forest plot showing the pooled odds ratio of the association between availability of standard guideline and utilization of routine health information health workers in Ethiopia, 2020.**

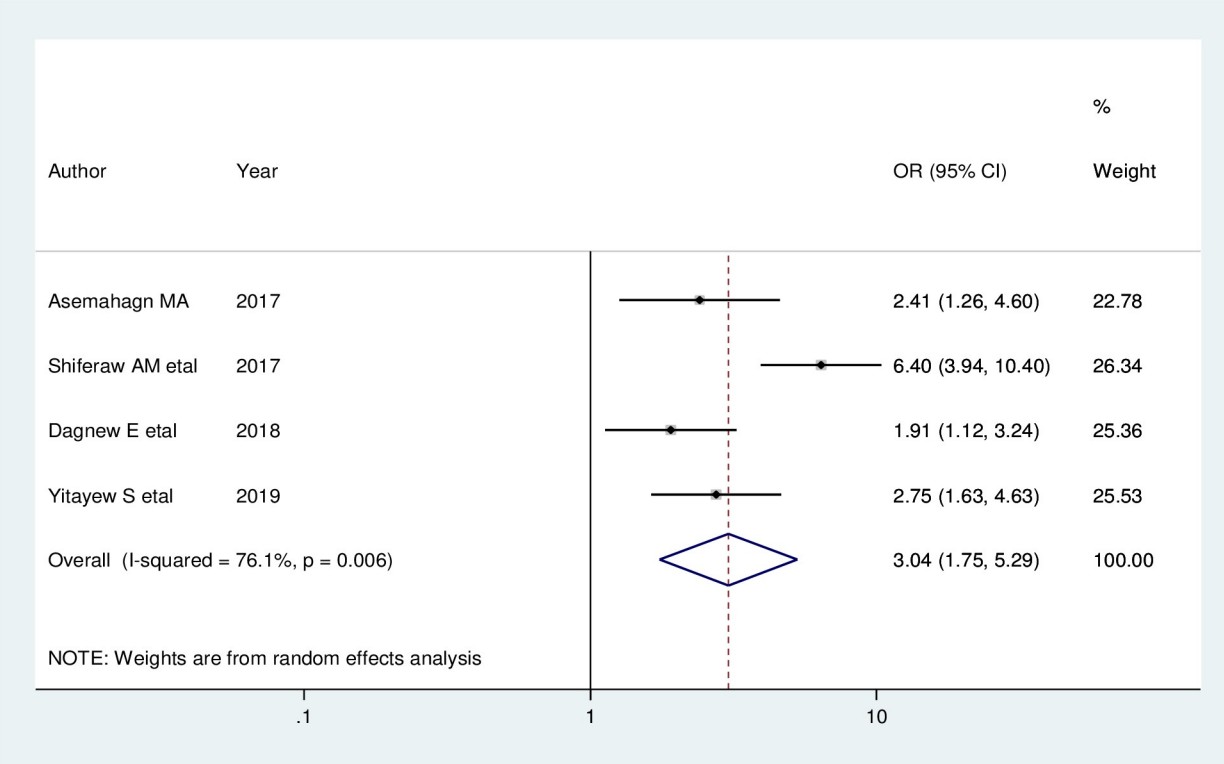

**Fig 8. Forest plot showing the pooled odds ratio of the association between data management knowledge and utilization of routine health information health workers in Ethiopia, 2020.**

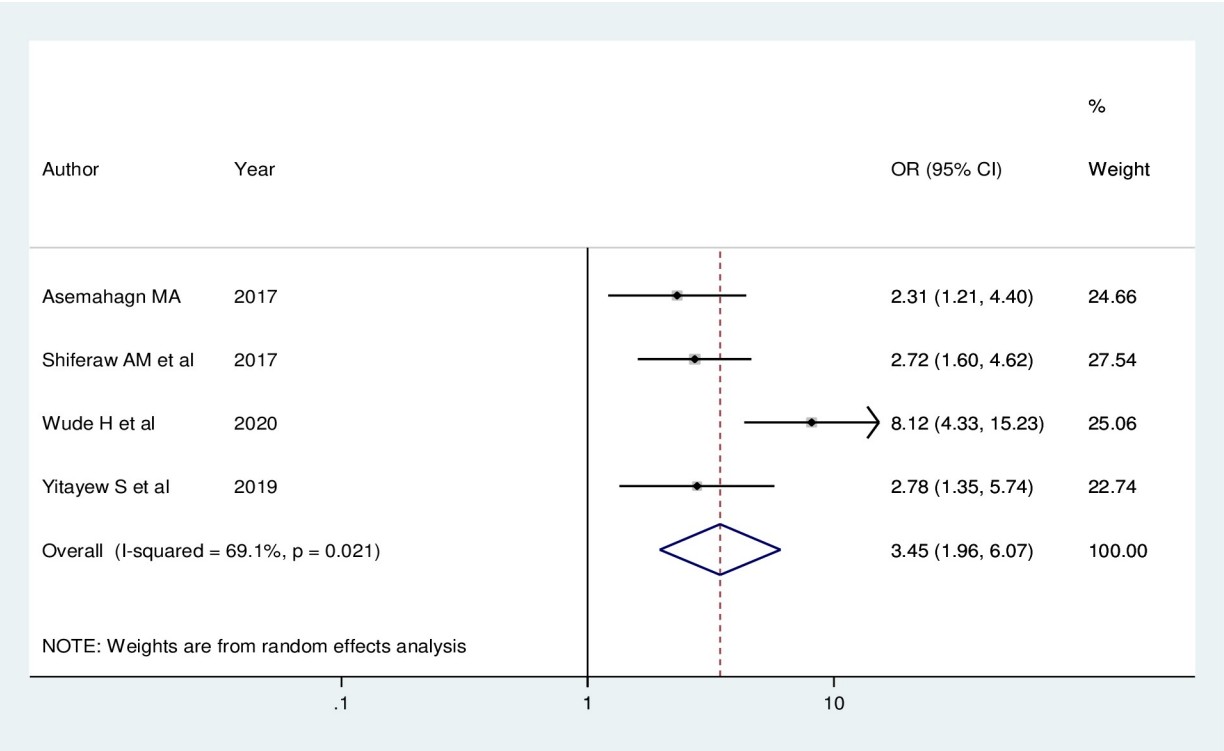

**Fig 9. Forest plot showing the pooled odds ratio of the association between training on health information and utilization of routine health information health workers in Ethiopia, 2020.**

trained on health information can have the potential to collect, compile, analyze, and utilize health information generated in the routine day-to-day activities. Moreover, workshop presentations are one of the outputs of data processing skills which will then increase health workers' data management knowledge and utilization of routine health information.

## Limitations of the study

Though this study is the first systematic review and meta-analysis about routine health information utilization among health workers in Ethiopia, it was not without limitations. In this meta-analysis, articles published only in the English language and have available full-text versions were included. The pooled odds ratio for all variables associated with routine health information utilization among health workers were not examined because the included studies classified the variables in different ways. All of the included articles were facility based cross-sectional studies which may reduce the generalizability of the finding. Furthermore, this study represented only studies reported from four regions which may affect the pooled prevalence of routine health information utilization.

## Conclusion

This systematic review and meta-analysis found that more than two-fifth of health workers did not use their routine health information. Supportive supervision, regular feedback, availability of standard guideline, data management knowledge and training on health information were identified factors associated with utilization of routine health information among health workers in Ethiopia. This study suggests the need to conduct regular supportive supervision,

provision of training and capacity building, mentoring on competence of routine health information tasks, and strengthening regular feedback at all health facilities with collaborative effort of policy-makers, programmers, and implementers as well as other concerned stakeholders. In addition, improving the accessibility and availability of standard set of indicators (guidelines)is important to scale-up information use.

## Supporting information

**S1 Checklist. Preferred Reporting Items for Systematic Reviews and Meta-Analyses (PRISMA) checklist.**
(DOC)

## Acknowledgments

The authors would like to thank the authors of the included primary studies, which used as source of information to conduct this systematic review and meta-analysis.

## Author Contributions

**Conceptualization:** Birye Dessalegn Mekonnen.

**Data curation:** Birye Dessalegn Mekonnen, Senafekesh Biruk Gebeyehu.

**Formal analysis:** Birye Dessalegn Mekonnen, Senafekesh Biruk Gebeyehu.

**Funding acquisition:** Birye Dessalegn Mekonnen.

**Investigation:** Birye Dessalegn Mekonnen.

**Methodology:** Birye Dessalegn Mekonnen, Senafekesh Biruk Gebeyehu.

**Supervision:** Senafekesh Biruk Gebeyehu.

**Validation:** Birye Dessalegn Mekonnen, Senafekesh Biruk Gebeyehu.

**Visualization:** Birye Dessalegn Mekonnen.

**Writing – original draft:** Birye Dessalegn Mekonnen, Senafekesh Biruk Gebeyehu.

**Writing – review & editing:** Birye Dessalegn Mekonnen, Senafekesh Biruk Gebeyehu.

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
