## [Decision Letter · Decision Letter 0]

11 Apr 2021

PONE-D-21-01653

Routine health information utilization and associated factors among health care workers in Ethiopia: a systematic review and meta-analysis

PLOS ONE

Dear Dr. Mekonnen,

Thank you for submitting your manuscript to PLOS ONE. After careful consideration, we feel that it has merit but does not fully meet PLOS ONE’s publication criteria as it currently stands. Therefore, we invite you to submit a revised version of the manuscript that addresses the points raised during the review process.

We look forward to receiving your revised manuscript.

Kind regards,

Frank T. Spradley

Academic Editor

PLOS ONE

We note that you have stated that you did not include date limits on your search for articles. Yet, your search results included studies from 2011-2020. Please amend your Methods section and results to more clearly define the date range that was searched, and if applicable, explain why no studies earlier than 2011 were included.

We note that this manuscript is a systematic review or meta-analysis; our author guidelines therefore require that you use PRISMA guidance to help improve reporting quality of this type of study. Please upload copies of the completed PRISMA checklist as Supporting Information with a file name “PRISMA checklist”.

In your Data Availability statement, you have not specified where the minimal data set underlying the results described in your manuscript can be found. PLOS defines a study's minimal data set as the underlying data used to reach the conclusions drawn in the manuscript and any additional data required to replicate the reported study findings in their entirety. All PLOS journals require that the minimal data set be made fully available. For more information about our data policy, please see http://journals.plos.org/plosone/s/data-availability.

Please include captions for your Supporting Information files at the end of your manuscript, and update any in-text citations to match accordingly. Please see our Supporting Information guidelines for more information: http://journals.plos.org/plosone/s/supporting-information.

Reviewers' comments:

Reviewer's Responses to Questions

**Comments to the Author**

1. Is the manuscript technically sound, and do the data support the conclusions?

Reviewer #1: Yes

Reviewer #2: Yes

2. Has the statistical analysis been performed appropriately and rigorously? 

Reviewer #1: I Don't Know

Reviewer #2: Yes

3. Have the authors made all data underlying the findings in their manuscript fully available?

Reviewer #1: Yes

Reviewer #2: Yes

4. Is the manuscript presented in an intelligible fashion and written in standard English?

Reviewer #1: Yes

Reviewer #2: No

5. Review Comments to the Author

Reviewer #1: 1.The authors said that "Health information is the processed and generated data that an individual, group or institutions use to support their decisions in the health care system." It's still confused me about the precise definition of Routine health information utilization. Do participants only need to read patients' medical records? Or do some statistical analysis and report their conclusion?

2.How to make a quantitative assessment of the utilization of routine health information? The authors said that "computed by dividing the number of health workers who had a good level of routine health information utilization by the total number of health workers in the study (sample size) multiplied by 100." So what is the standard of "good level of routine health information utilization"? Notably, whether the standards in 10 studies included in this meta-analysis papers are consistent?

3.I noticed all identified factors associated with Routine health information utilization were personal factors. I understand that healthcare workers use this routine health information during clinical work. However, the same person in a different objective environment has a different ability to use routine health information. For instance, if I worked in a hospital with poor informatization, I might have to write medical records by hand and had poor health information utilization. In 10 studies included in this meta-analysis, differences in hospitals will be a big bias factor.

4.It is emphasized that this is the first report in Ethiopia, but what is special about routine health information utilization of healthcare workers in Ethiopia? Are there some different factors compared with other countries?

5.The authors said that "In this meta-analysis, articles published only in the English language." Why only English publication if you emphasize the first report in Ethiopia?

Reviewer #2: This paper addresses the estimated impact of factors designed to facilitate use of routine health data on the utilization of the data in Ethiopia. This paper addresses the impact of factors designed to facilitate use of routine health data in Ethiopia and indeed in all developing countries. It does this through a meta-analysis of available and relevant data. This is an important issue. Improving the utilization of routine health data for decision-making at every level of the health systems is key to developing efficient health systems, and is especially important in countries where the funding for the system is limited.

The study uses appropriate methods to search for information, to extract the data, to assess bias and to select and measure the outcomes. It is described clearly and with detail in the Methods section. I am unfamiliar with the Funnel and Forest plots so cannot comment on these.

There is an extensive review of literature on the District Health System in other African countries which provides the context for the aims and results of the present investigation.

The outcomes noted that training, supervision and support for health workers, and regular feedback is associated with improved utilization of health data. The existence of guidelines and data management training also has a positive impact. The limitations are clearly stated. The recommendations arising out of these findings should encourage health systems managers in developing countries to further implement the measures in order to make better use of their information systems.

There are many grammatical errors in the paper and it will be advisable for the authors should arrange to have these corrected.

6. PLOS authors have the option to publish the peer review history of their article (what does this mean?). If published, this will include your full peer review and any attached files.

Reviewer #1: **Yes: **Zhenghao Wu

Reviewer #2: No

---

## [Author Response · Author response to Decision Letter 0]

12 May 2021

Author’s Point-by-point response letter to reviewer

Title: Routine health information utilization and associated factors among health care workers in Ethiopia: a systematic review and meta-analysis

Authors: Birye Dessalegn Mekonnen and Senafekesh Biruk Gebeyehu

Corresponding author: Birye Dessalegn Mekonnen

Email: birye22@gmail.com

ORCID: 0000-0003-3879-1330

Teda Health Science College, Ethiopia 

May, 2021 

Dear, Editors of PLOS ONE

This is a point-by-point response letter that accompanies the responses for the editor and reviewers’ comments concerning the manuscript entitled ‘Routine health information utilization and associated factors among health care workers in Ethiopia: a systematic review and meta-analysis’. It is known that the manuscript has been reviewed by reviewers and sent back to authors to carry out the corrections to meet the reviewers’ concern, and resubmission. 

As authors of this manuscript, the comments and concerns raised by the reviewers’ were highly insightful and enabled us to improve the quality & scientific plausibility of the manuscript. To do so, we have tried to address all the editor’s and reviewers’ concerns point by point and described below in table as per your guide. Therefore, we are pleased to resubmit the revised version of the manuscript for further process and facilitation of its publication on PLOS ONE.

Author’s Point-by-point response

Reviewers’ Comments and authors’ responses

 Authors’ response: First of all, thank you very much for constructive feedback. We correct and ensure that our manuscript meets PLOS ONE's style requirements, including those for file naming.

2. We note that you have stated that you did not include date limits on your search for articles. Yet, your search results included studies from 2011-2020. Please amend your Methods section and results to more clearly define the date range that was searched, and if applicable, explain why no studies earlier than 2011 were included.

Authors’ response: First of all, thank you very much for constructive feedback. We tried to make it clear and correct in the document. All publications reported up to November 18, 2020 were considered. However, there were no studies reported earlier than 2011. Therefore, all the included studies were published and reported from 2011-2020. 

3. We note that this manuscript is a systematic review or meta-analysis; our author guidelines therefore require that you use PRISMA guidance to help improve reporting quality of this type of study. Please upload copies of the completed PRISMA checklist as Supporting Information with a file name “PRISMA checklist”.

Authors’ response: First of all, thank you very much for constructive feedback. We upload copies of the completed PRISMA checklist as Supporting Information with a file name “S1 Table” as per PLOS ONE's guidelines.

Authors’ response: First of all, thank you very much for constructive feedback. We will do that as per your requirement.

Authors’ response: First of all, thank you very much for constructive feedback. We include captions for our Supporting Information files at the end of your manuscript.

Reviewer #1: 

1. The authors said that "Health information is the processed and generated data that an individual, group or institutions use to support their decisions in the health care system." It's still confused me about the precise definition of Routine health information utilization. Do participants only need to read patients' medical records? Or do some statistical analysis and report their conclusion?

Authors’ response: Dear reviewer, first of all thank you very much for your interest for reviewing our paper, and constructive feedback. We tried to make more precise. The definition of ‘Health information’ and ‘Routine health information utilization’ is quite different. Yes, ‘Health information is the processed and generated data that an individual, group or institutions use to support their decisions in the health care system’. But, the operational definition for Routine health information utilization is included in the revised manuscript from the method section at the sub-topic of ‘outcome measure’. 

2.How to make a quantitative assessment of the utilization of routine health information? The authors said that "computed by dividing the number of health workers who had a good level of routine health information utilization by the total number of health workers in the study (sample size) multiplied by 100." So what is the standard of "good level of routine health information utilization"? Notably, whether the standards in 10 studies included in this meta-analysis papers are consistent?

Authors’ response: Dear reviewer, again thank you very much for your concern and constructive feedback. In the primary studies, utilization of routine health information was assessed using a standardized assessment tool which adapted from the Performance of Routine Information System Management (PRISM) framework tool. In this systematic review and meta-analysis, the quantitative assessment of the utilization of routine health information was computed from the reported good level of routine health information utilization in the primary studies against the total number of health workers included in the study.

3.I noticed all identified factors associated with Routine health information utilization were personal factors. I understand that healthcare workers use this routine health information during clinical work. However, the same person in a different objective environment has a different ability to use routine health information. For instance, if I worked in a hospital with poor informatization, I might have to write medical records by hand and had poor health information utilization. In 10 studies included in this meta-analysis, differences in hospitals will be a big bias factor.

Authors’ response: Dear reviewer, again thank you very much for your concern and constructive feedback. Of course, the same person in a different objective environment may have a different ability to use routine health information. This differences in hospitals may be a bias factor. This systematic review and meta-analysis computed the pooled estimate of determinants which were reported in the primary studies. However, because of primary studies have not investigated facility level factors, almost all identified factors associated with Routine health information utilization were personal factors.

4.It is emphasized that this is the first report in Ethiopia, but what is special about routine health information utilization of healthcare workers in Ethiopia? Are there some different factors compared with other countries?

Authors’ response: Dear reviewer, again thank you very much for your concern and constructive feedback. Sorry, we are not clear for the question you raised ‘what is special about routine health information utilization of healthcare workers in Ethiopia?’ In this study, identified factors associated with utilization of routine health information were also supported with other studies conducted in some other countries. However, some different factors have identified in this study compared with other countries.

5.The authors said that "In this meta-analysis, articles published only in the English language." Why only English publication if you emphasize the first report in Ethiopia?

Authors’ response: Dear reviewer, again thank you very much for your concern and constructive feedback. To the authors knowledge, this study is the first systematic review and meta-analysis about routine health information utilization among health workers in Ethiopia. Again, we don’t understand your concern that you relate ‘only English publication’ and ‘being the first report in Ethiopia’. In the study setting (Ethiopia), the possibility of studies conducted and reported in the English language is very less likely. Moreover, if it isn’t more necessary to state it as limitation of study we can remove it. 

Reviewer #2: 

This paper addresses the estimated impact of factors designed to facilitate use of routine health data on the utilization of the data in Ethiopia. This paper addresses the impact of factors designed to facilitate use of routine health data in Ethiopia and indeed in all developing countries. It does this through a meta-analysis of available and relevant data. This is an important issue. Improving the utilization of routine health data for decision-making at every level of the health systems is key to developing efficient health systems, and is especially important in countries where the funding for the system is limited.

Authors’ response: Dear reviewer, first of all thank you very much for your interest for reviewing our paper, and constructive feedback. We appreciate your insight regarding our paper. 

The study uses appropriate methods to search for information, to extract the data, to assess bias and to select and measure the outcomes. It is described clearly and with detail in the Methods section. I am unfamiliar with the Funnel and Forest plots so cannot comment on these.

Authors’ response: Dear reviewer, again thank you very much for your concern and insight.

There is an extensive review of literature on the District Health System in other African countries which provides the context for the aims and results of the present investigation.

Authors’ response: Dear reviewer, again thank you very much for your concern and insight.

The outcomes noted that training, supervision and support for health workers, and regular feedback is associated with improved utilization of health data. The existence of guidelines and data management training also has a positive impact. The limitations are clearly stated. The recommendations arising out of these findings should encourage health systems managers in developing countries to further implement the measures in order to make better use of their information systems.

Authors’ response: Dear reviewer, again thank you very much for your concern and constructive feedback. We really appreciate the way you see and understand the full manuscript. 

There are many grammatical errors in the paper and it will be advisable for the authors should arrange to have these corrected.

Author’s response: Dear reviewer, again thank you very much for your constructive feedback and concern. We accept and tried to modify it in the manuscript. We made grammatical corrections as per your concern.

I look forward to hear from you at your earliest convenience.

Birye Dessalegn Mekonnen (MPH in Reproductive and Child Health)

Corresponding author

---

## [Decision Letter · Decision Letter 1]

8 Jun 2021

PONE-D-21-01653R1

Routine health information utilization and associated factors among health care workers in Ethiopia: a systematic review and meta-analysis

PLOS ONE

Dear Dr. Mekonnen,

Thank you for submitting your manuscript to PLOS ONE. After careful consideration, we feel that it has merit but does not fully meet PLOS ONE’s publication criteria as it currently stands. Therefore, we invite you to submit a revised version of the manuscript that addresses the points raised during the review process.

The reviewer still has some comments that must be addressed. Notably, the authors must contact a copyeditor to proof the English grammar and syntax prior to resubmitting this manuscript.

A rebuttal letter that responds to each point raised by the academic editor and reviewer(s). You should upload this letter as a separate file labeled 'Response to Reviewers'.A marked-up copy of your manuscript that highlights changes made to the original version. You should upload this as a separate file labeled 'Revised Manuscript with Track Changes'.An unmarked version of your revised paper without tracked changes. You should upload this as a separate file labeled 'Manuscript'

We look forward to receiving your revised manuscript.

Kind regards,

Frank T. Spradley

Academic Editor

PLOS ONE

Additional Editor Comments (if provided):

Reviewers' comments:

Reviewer's Responses to Questions

**Comments to the Author**

1. If the authors have adequately addressed your comments raised in a previous round of review and you feel that this manuscript is now acceptable for publication, you may indicate that here to bypass the “Comments to the Author” section, enter your conflict of interest statement in the “Confidential to Editor” section, and submit your "Accept" recommendation.

Reviewer #2: (No Response)

2. Is the manuscript technically sound, and do the data support the conclusions?

Reviewer #2: Yes

3. Has the statistical analysis been performed appropriately and rigorously? 

Reviewer #2: Yes

4. Have the authors made all data underlying the findings in their manuscript fully available?

Reviewer #2: Yes

5. Is the manuscript presented in an intelligible fashion and written in standard English?

Reviewer #2: No

6. Review Comments to the Author

Reviewer #2: The article still contains many minor grammatical errors. For example, In the first sentence of the Introduction the authors state "Health information is the processed and generated data that an individual, group or institutionS use . . " mixing singular and plural amongst the nouns and verbs. While the meaning is clear the format is incorrect. Other examples: "A proper[ly] functioning health information system" and " "data usually sat in [on] the shelves. These minor errors do not affect the meaning of the article, but because PLOS does not copy-edit articles it is important the the authors attend to this aspect of presentation.

In my view the authors have responded adequately to the other comments from the reviewers. The findings from this meta-analysis of 10 studies in Ethiopia show that very simple measures such as good training of health workers, supportive supervision and regular feedback can improve the uptake of routine health data substantially. The problem of poor uptake of information is a common one in developing countries and the authors point to similar findings in several African countries. The article is timely, not only for African health systems, but globally. The COVID-19 pandemic has made us all aware that health systems across the world are interdependent, and should all function efficiently in order to contain communicable diseases which may spread from country to country. I would therefore urge the authors to attend to the copy-editing problem so as not to delay publication.

7. PLOS authors have the option to publish the peer review history of their article (what does this mean?). If published, this will include your full peer review and any attached files.

Reviewer #2: No

---

## [Author Response · Author response to Decision Letter 1]

16 Jun 2021

Author’s Point-by-point response letter to reviewer

Title: Routine health information utilization and associated factors among health care workers in Ethiopia: a systematic review and meta-analysis

Authors: Birye Dessalegn Mekonnen and Senafekesh Biruk Gebeyehu

Corresponding author: Birye Dessalegn Mekonnen

Email: birye22@gmail.com

ORCID: 0000-0003-3879-1330

Teda Health Science College, Ethiopia 

June, 2021 

Dear, Editors of PLOS ONE

This is a point-by-point response letter that accompanies the responses for the editor and reviewers’ comments concerning the manuscript entitled ‘Routine health information utilization and associated factors among health care workers in Ethiopia: a systematic review and meta-analysis’. It is known that the manuscript has been reviewed by reviewers and sent back to authors to carry out the corrections to meet the reviewers’ concern, and resubmission for the second time. 

As authors of this manuscript, the comments and concerns raised by the reviewers’ were highly insightful and enabled us to improve the quality & scientific plausibility of the manuscript. To do so, we have tried to address all the editor’s and reviewers’ concerns point by point and described below in table as per your guide. Therefore, we are pleased to resubmit the revised version of the manuscript for further process and facilitation of its publication on PLOS ONE. Here below are Author’s Point-by-point responses.

Author’s Point-by-point response

Reviewers’ Comments and authors’ responses

Editor’s concern 1:

The reviewer still has some comments that must be addressed. Notably, the authors must contact a copyeditor to proof the English grammar and syntax prior to resubmitting this manuscript.

Author’s response: Dear Editor, first of all thank you very much for your constructive feedback and concern. We have carefully review and made some amendments to the grammatical structure of the manuscript. We have also invited English language expertise to review this manuscript, and they made some corrections. Thus, we made grammatical corrections as per your concern.

Additional Editor Comments (if provided):

Author’s response: Dear Editor, again thank you very much for your concern. We have checked our reference list to ensure that it is complete and correct. We haven’t cited papers that have been retracted. Thus, we didn’t make any change in the reference list.

Reviewer #2: 

The article still contains many minor grammatical errors. For example, In the first sentence of the Introduction the authors state "Health information is the processed and generated data that an individual, group or institutionS use . . " mixing singular and plural amongst the nouns and verbs. While the meaning is clear the format is incorrect. Other examples: "A proper[ly] functioning health information system" and " "data usually sat in [on] the shelves. These minor errors do not affect the meaning of the article, but because PLOS does not copy-edit articles it is important the the authors attend to this aspect of presentation.

Authors’ response: Dear reviewer, thank you very much for your interest for reviewing our paper, and constructive feedback. We accept your concern and tried to made some modification in the manuscript. We have carefully review and made some amendments to the grammatical structure of the manuscript as per your concern.

In my view the authors have responded adequately to the other comments from the reviewers. The findings from this meta-analysis of 10 studies in Ethiopia show that very simple measures such as good training of health workers, supportive supervision and regular feedback can improve the uptake of routine health data substantially. The problem of poor uptake of information is a common one in developing countries and the authors point to similar findings in several African countries. The article is timely, not only for African health systems, but globally. The COVID-19 pandemic has made us all aware that health systems across the world are interdependent, and should all function efficiently in order to contain communicable diseases which may spread from country to country. I would therefore urge the authors to attend to the copy-editing problem so as not to delay publication.

Authors’ response: Dear reviewer, again thank you very much for your interest for reviewing our paper, and constructive feedback.

I look forward to hear from you at your earliest convenience.

Birye Dessalegn Mekonnen (MPH in Reproductive and Child Health)

Corresponding author

---

## [Editor Report · Decision Letter 2]

23 Jun 2021

Routine health information utilization and associated factors among health care workers in Ethiopia: a systematic review and meta-analysis

PONE-D-21-01653R2

Dear Dr. Mekonnen,

We’re pleased to inform you that your manuscript has been judged scientifically suitable for publication and will be formally accepted for publication once it meets all outstanding technical requirements.

Kind regards,

Frank T. Spradley

Academic Editor

PLOS ONE

---

## [Editor Report · Acceptance letter]

25 Jun 2021

PONE-D-21-01653R2 

Routine health information utilization and associated factors among health care workers in Ethiopia: a systematic review and meta-analysis 

Dear Dr. Mekonnen:

I'm pleased to inform you that your manuscript has been deemed suitable for publication in PLOS ONE. Congratulations! Your manuscript is now with our production department. 

Kind regards, 

on behalf of

Dr. Frank T. Spradley 

Academic Editor

PLOS ONE